# Praziquantel and risk of visual disorders: Case series assessment

**Merhawi Debesai**  *, **Mulugeta Russom**

Eritrean Pharmacovigilance Centre, National Medicines and Food Administration, Asmara, Eritrea

* dome.bable07@gmail.com

## Abstract

### Introduction

Praziquantel has been in use by helminthiasis and schistosomiasis control programs for about 30 years. Although deemed to be safe with regard to its adverse drug reaction profile in reference to the product information of Biltricide, the Eritrean Pharmacovigilance Center received reports of visual abnormalities related to the drug. This is a case series assessment of unusual cases of visual abnormalities associated with praziquantel.

### Methods

Search was made in VigiBase by setting praziquantel as a drug substance, Eritrea as the reporting country and all eye disorders, high level term (HLT) to capture all visual disorders associated with Praziquantel. The retrieved dataset was exported into an Excel spreadsheet for descriptive analysis and causality was assessed using Austin Bradford-Hill criteria.

### Results

There are a total of 2579 Individual Case Safety Reports (ICSRs) of various Adverse Drug Reactions (ADRs) of praziquantel reported from Eritrea in VigiBase. The 61 reports of visual abnormalities that arose within the first 24 hours of praziquantel administration are of note. With a strong association as evidenced by the positive $IC_{025}$ value, the association of praziquantel and blurred vision was consistently reported from different health facilities over a period of three years. It is a specific association in terms of both the exposure (only praziquantel) and the outcome (blurred vision) as reported in majority of the cases. However, experimental evidences for the association are lacking, the underweight profile of the Eritrean population suggests overdosing as a possible risk factor for the occurrence of these reactions.

### Conclusion

The strength, temporal plausibility, consistency and specificity of the association are suggestive of a causal association between praziquantel and visual disorders.

**Data Availability Statement:** The data underlying the results presented in the study are available from VigiBase, the single largest drug safety data repository in the world, for researchers who meet

the criteria for access to confidential data via the following address: Vigibase@who-umc.org

**Funding:** The author(s) received no specific funding for this work.

**Competing interests:** The authors have declared that no competing interests exist.

## Author summary

The Eritrean Pharmacovigilance centre, since its integration in Mass Drug Administration in 2017, has been receiving large proportion of reports of adverse drug reactions associated with the drug praziquantel administered in mass drug campaigns. Of note were reports of the visual abnormalities namely blurred vision and visual impairment which are not known to be related to the drug. On a probability assessment, the series of 61 cases seems to show a reasonable link between the reported adverse reactions and praziquantel but not an absolute cause and effect relationship. This study suggests inaccuracies in the dose of praziquantel, in reference to some African studies, as a possible root cause, that might have led to overdosing of some subjects and hence the occurrence of the reported reactions in noticeable rates. It also addresses primarily a public health issue, rendering the findings interesting to both the scientific and non-scientific communities in general and disease control programs in particular. It is a driving force to conduct further research, improve our understanding of praziquantel and its safety profile, and the use of the drug in better ways.

## Introduction

Drugs like Praziquantel have been widely used for Soil-Transmitted helminthiasis and schistosomiasis control as those infections are burdensome to school-aged children in many parts of the world [1]. Praziquantel, a broad spectrum anthelmintic with activity against trematode or cestode helminthic infections of human and veterinary origin, has been in use since 1980 [2]. The dose of praziquantel in Mass Drug Administration (MDA) continues to be calculated based on the height of the recipient in reference to the WHO dose pole [3,4] in many countries around the world including Eritrea. Studies from some African countries however demonstrate marked inaccuracies in the WHO dosing recommendations and suggest adopting their own pole [5,6,7].

The Eritrean Pharmacovigilance Center has been working relentlessly to integrate its activities in various public health programs. One of these is the Neglected Tropical Diseases (NTD) control program. As a result of the collaboration between the Eritrean Pharmacovigilance Center and NTD control program, stimulated passive surveillance activities in Mass Drug Campaigns were partially implemented in 2017. The outcome was a surge of reports of adverse events related primarily to Praziquantel use in MDA. In September 2019, there were close to 2600 reports of Praziquantel related adverse events in the WHO global individual case safety reports (ICSRs) database, VigiBase [8] submitted from Eritrea. These represents close to 65% of the globally reported cases to this drug. The reactions comprised largely various gastrointestinal disorders as nausea and vomiting and other systemic events.

Of note were the 61 cases of visual abnormalities namely blurred vision and visual impairments observed in several cases most of which were received in 2017. The product information of Biltricide (praziquantel 600mg tablets) does not mention visual problems in the adverse effects section of the document [9] nor does other references except for post-marketing surveillance reports of visual disturbance in one drug information website [10]. This work is therefore to assess the causal relationship of visual abnormalities and Praziquantel.

## Methods

On September 28, 2019, data was retrieved from VigiBase, the WHO international ICSRs database developed and maintained by the Uppsala Monitoring Center (UMC), Sweden, via VigiLyze; an analysis tool also developed and maintained by the UMC.

First, search was made to identify all praziquantel related adverse events by setting 'Praziquantel' as drug substance and 'all adverse events' as reactions term. This provided us reports of praziquantel related adverse events at global and country level. As the aim of the study was to the assess causal relationship of praziquantel and visual disorders, the search was further narrowed down by applying 'Eritrea' as country, 'praziquantel' as drug substance and 'eye disorders' (system organ class), as reaction term. The retrieved dataset was then exported into Excel spreadsheet for descriptive analysis (mean, median and frequency measures). Causality assessment in case series was done using Austin Bradford-Hill criteria [11] by evaluating the available information on the association of praziquantel with blurred vision against the nine criteria namely the association's strength, temporality, specificity, consistency, biological plausibility, dose response relationship, experimental evidence, available analogies and coherence with established science.

In all ICSRs retrieved from VigiBase, the identities of individual cases are anonymous, thus; individual level information is held confidential.

## Results

A total of 3968 adverse events related to praziquantel, submitted from different countries, were retrieved from the WHO global ICSRs database. Of these adverse events, 65% (n = 2579), from which sixty-one cases of visual abnormalities associated with praziquantel namely blurred vision (47), visual impairment (13) and lacrimation increased (1) were retrieved, were submitted by Eritrea. (Fig 1).

All the 61 cases were reported in the time period between January 2017 and August 2019. The reports came with a mean completeness score of 0.85 in the scale of 0 to 1. The Information Component ($IC_{025}$) generated by the database for the reaction terms blurred vision and visual impairment was +0.82 and -1.01 respectively.

The age of the recipient was reported in all but one case and the median age was found to be 33 years (range 10–72). Cases comprised a similar sex distribution with more females (35/

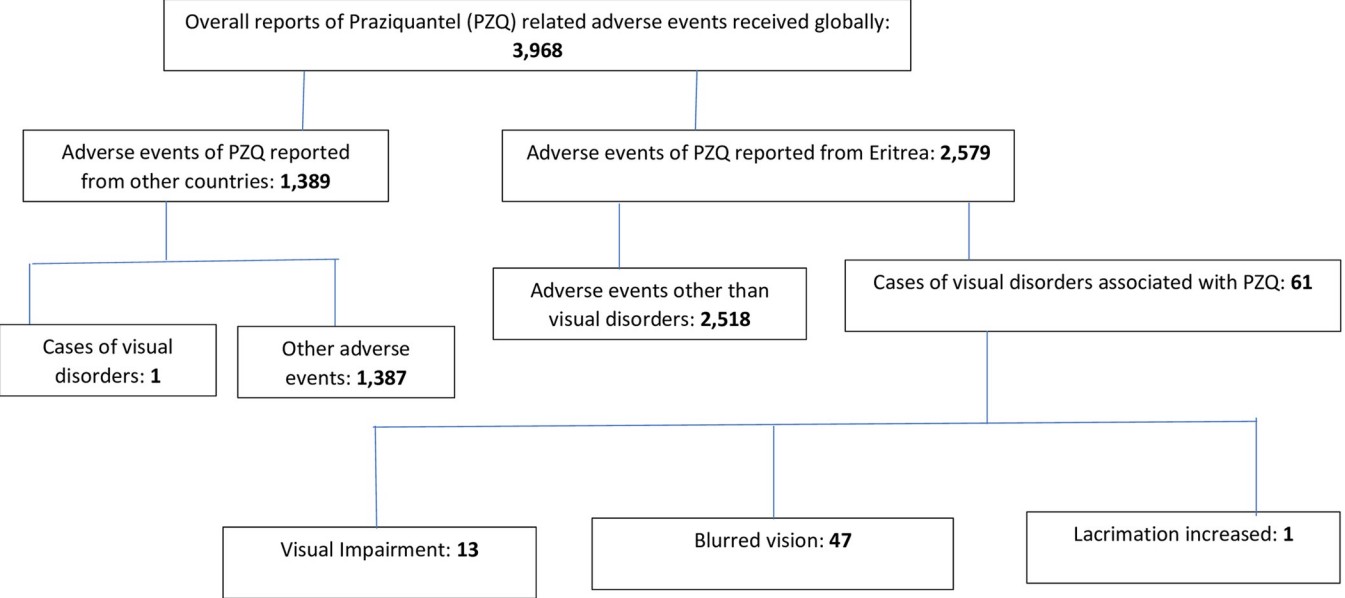

**Fig 1. Summary results of Praziquantel related cases of visual abnormalities submitted from Eritrea to the WHO global individual case safety reports database.**

61) than males. The time to reaction onset as reported in the 61 cases lies within the same day of administration of the drug.

Praziquantel was the only suspected drug in 47 of the 61 cases, in the rest 13 cases Mebendazole was co-reported (as a co-suspect in seven and concomitant in six cases) while in the remaining one case, Albendazole was a co-suspect. In 28 of the 61 cases blurred vision (26) and visual impairment (2) were the sole reported reactions. The remaining 33 cases reported additional reactions including GI-disorders, dizziness, headache and other systemic side effects. The dosing variation ranged from 1 tablet (600mg) to 4 tablets (2400mg) of praziquantel while the dose of Menbendazole was 500mg in all the 13 cases in which the drug is co-reported. Table 1 below summarizes some key variables of each of the 61 cases retrieved. Praziquantel was administered as a single dose only in all the cases for prophylaxis in MDA and thus, information on de-challenge and re-challenge were unobtainable. All the 61 events were reported to have been resolved spontaneously soon after the onset.

## Discussion

The positive IC value of praziquantel and blurred vision in the WHO global ICSRs database indicates that visual abnormalities were commonly reported than expected; making the association strong. The association of Praziquantel and visual abnormalities has been consistently reported over a period of three years although about 80% of the reports were received in the first year alone. This difference may be attributed to variation in the surveillance initiatives and coverage of MDA over the three years. There was also consistency in time to reaction onset, that is, reactions observed within the first 24 hours following administration of praziquantel in all the cases and all reactions resolved shortly following the onset. The association was also specific as visual abnormalities were the only reported reaction terms following the sole administration of praziquantel in 47.5% of the cases. The occurrence of the reactions within the first 24 hours following the administration of praziquantel in all the cases suggests a plausible temporal association.

To the best of the authors knowledge, there were however no experimental evidences from pre-clinical or clinical trials that associate visual abnormalities with Praziquantel. Evidence suggesting specific dose-response relationship between Praziquantel and the visual abnormalities are lacking at the moment. The single dose administered to the cases ranges from 1–4 tablets (600mg - 2400mg) and it was determined by referring to the WHO height-based dose pole for Praziquantel [3,4]. According to the data from the National Statistics Office of Eritrea, more than a half of the Eritrean adult population was found to be underweight in reference to the age for weight band records of various age groups (in the range 16 to 50 years) compared to the global age for weight band suggested by the WHO. In this regard, as mass drug administration of praziquantel uses height-based dosing in Eritrea, some patients might be exposed to overdose of praziquantel which might be a risk factor, among others, for the frequent reports of visual abnormalities. Inaccuracies in the WHO dose pole for dose calculation of Praziquantel were clearly demonstrated in some studies from countries in Sub-Saharan Africa [5,6,7]. These studies recommend countries to customize the WHO dose pole in order to fit their population's situation.

The deployment of mass drug administration of Praziquantel and implementation of stimulated passive surveillance of adverse events in Eritrea led to increased incidences of reporting of various drug related adverse events. Thousands of complaints following the administration of Praziquantel in mass campaigns might have been reported to health professionals. It is only in recent years that such complaints were appropriately documented and reported to the Eritrean Pharmacovigilance Center. About 2600 reports dominated by the common

**Table 1. Summary of selected variables for the 61 cases of visual abnormalities related to praziquantel (PZQ).**

| Case No. | Sex | Age (Y) | Co-reported Drugs: Suspect (S) or Concomitant(C) | PZQ Dose | Reaction(s) |
|---|---|---|---|---|---|
| 1 | Male | - | Mebendazole (S) | 1200 mg | Vision blurred |
| 2 | Male | 10 | Mebendazole (S) | 1200 mg | Vision blurred |
| 3 | Female | 11 | None | 1500 mg | Vision blurred |
| 4 | Female | 11 | None | 1200 mg | Visual impairment, nausea and headache |
| 5 | Male | 12 | None | 1200 mg | Visual impairment and GI-disorder |
| 6 | Male | 12 | None | 1200 mg | Vision blurred |
| 7 | Female | 12 | Mebendazole (S) | 1500 mg | Vision blurred |
| 8 | Female | 13 | None | 1500 mg | Vision blurred |
| 9 | Male | 14 | Mebendazole (C) | 1500 mg | Vision blurred and headache |
| 10 | Male | 14 | None | 2400 mg | Vision blurred |
| 11 | Male | 14 | None | 1200 mg | Visual impairment, headache and GI-disorder |
| 12 | Male | 14 | Mebendazole (S) | 1200 mg | Vision blurred |
| 13 | Male | 15 | Mebendazole (S) | 2400 mg | Vision blurred, malaise and balance disorder |
| 14 | Male | 15 | None | 1200 mg | Vision blurred |
| 15 | Female | 15 | Mebendazole (C) | - | Vision blurred, nausea, dizziness and asthenia |
| 16 | Female | 15 | None | 1800 mg | Vision blurred |
| 17 | Male | 17 | None | 2400 mg | Visual impairment |
| 18 | Female | 18 | None | 1800 mg | Vision blurred |
| 19 | Female | 19 | None | 1800 mg | Visual impairment, headache and dyspepsia |
| 20 | Female | 19 | None | 2400 mg | Vision blurred, dizziness and abdominal pain |
| 21 | Female | 19 | None | 2400 mg | Vision blurred |
| 22 | Female | 19 | None | 1800 mg | Vision blurred |
| 23 | Female | 20 | Mebendazole (S) | 1200 mg | Vision blurred and vomiting |
| 24 | Female | 20 | None | 1500 mg | Vision blurred |
| 25 | Female | 22 | None | 2400 mg | Visual impairment, headache and GI-disorder |
| 26 | Female | 25 | Albendazole (S) | 1500 mg | Visual impairment, vomiting and vertigo |
| 27 | Male | 25 | None | 1800 mg | Vision blurred |
| 28 | Female | 27 | None | 1800 mg | Vision blurred |
| 29 | Male | 28 | None | 2400 mg | Visual impairment |
| 30 | Male | 32 | None | 1800 mg | Vision blurred |
| 31 | Female | 33 | None | 2400 mg | Vision blurred |
| 32 | Female | 33 | None | 2400 mg | Vision blurred |
| 33 | Male | 35 | None | 1800 mg | Vision blurred |
| 34 | Female | 38 | Mebendazole (C) | 600 mg | Lacrimation increased, headache, chest pain and nausea |
| 35 | Female | 38 | None | 1800 mg | Visual impairment, dyspepsia and GI-disorder |
| 36 | Female | 40 | None | 2400 mg | Visual impairment and vomiting |
| 37 | Female | 40 | None | 2400 mg | Vision blurred, abdominal pain and headache |
| 38 | Female | 40 | None | 1800 mg | Vision blurred |
| 39 | Male | 40 | None | 1800 mg | Vision blurred, dizziness and vomiting |
| 40 | Female | 45 | Mebendazole (C) | - | Vision blurred, nausea, asthenia and dizziness |
| 41 | Female | 45 | None | 1800 mg | Visual impairment, dyspepsia, nausea and headache |
| 42 | Male | 45 | Mebendazole (C) | - | Vision blurred, dizziness and headache |
| 43 | Female | 48 | None | 1800 mg | Vision blurred, dizziness and abdominal pain |
| 44 | Female | 48 | None | 2400 mg | Vision blurred |
| 45 | Female | 50 | None | 2400 mg | Vision blurred |
| 46 | Male | 50 | None | 2400 mg | Vision Blurred and dizziness |
| 47 | Female | 50 | None | 1800 mg | Vision blurred, dizziness and asthenia |

**Table 1.** (Continued)

| Case No. | Sex | Age (Y) | Co-reported Drugs: Suspect (S) or Concomitant(C) | PZQ Dose | Reaction(s) |
|---|---|---|---|---|---|
| 48 | Male | 50 | None | 2400 mg | Vision blurred |
| 49 | Female | 58 | None | 1800 mg | Vision blurred |
| 50 | Male | 60 | None | 2400 mg | Vision blurred |
| 51 | Female | 60 | None | 1800 mg | Visual impairment and GI-disorder |
| 52 | Male | 60 | None | 2400 mg | Vision blurred |
| 53 | Female | 60 | Mebendazole (S) | - | Vision blurred and headache |
| 54 | Female | 62 | None | 2400 mg | Vision blurred, dizziness and abdominal pain |
| 55 | Male | 63 | Mebendazole (C) | - | Vision blurred and headache |
| 56 | Male | 65 | None | 1800 mg | Vision blurred, asthenia and abdominal pain |
| 57 | Male | 65 | None | 2400 mg | Vision blurred, abdominal pain and vomiting |
| 58 | Male | 70 | None | 2400 mg | Vision blurred |
| 59 | Female | 70 | None | 2400 mg | Vision blurred and abdominal discomfort |
| 60 | Male | 71 | None | 2400 mg | Vision blurred and headache |
| 61 | Female | 72 | None | 2100 mg | Visual impairment and GI-disorder |

gastrointestinal side effects may not demonstrate an alarming situation as a whole taking into account the multiple campaigns carried out and hundreds of thousands of doses given in the same time period. Yet, the observation of several cases with visual abnormalities that are unexpected is certainly alarming.

This study had several limitations. Although Mebendazole 500mg tablet is normally co-administered with praziquantel in MDA, praziquantel was consistently reported as solely administered in several cases. Thus, exposure status might not have been appropriately reported. Moreover, the data set used in this assessment comprises reports of passive surveillance; thus, due to lack of denominator information we cannot determine the incidence rate of visual abnormalities associated with praziquantel.

In conclusion, taking into account the strength of the association, its consistency and specificity and the plausible time to onset, there seems to be a suggestive causal relationship between praziquantel and visual abnormalities. Overdosing of praziquantel might be a factor to contribute to the frequent occurrence of unusual adverse effects. The authors, therefore, recommend further vigilance by healthcare professionals and strengthening the integration of Pharmacovigilance into the NTD control programs to capture similar cases in future campaigns, safeguard consumers and strengthen this safety signal. We also recommend studies to validate the WHO dose pole accuracy for the Eritrean population as overdose might be a risk factor for the occurrence of these adverse effects.

## Author Contributions

**Conceptualization:** Merhawi Debesai, Mulugeta Russom.

**Data curation:** Merhawi Debesai.

**Investigation:** Merhawi Debesai, Mulugeta Russom.

**Methodology:** Mulugeta Russom.

**Writing – Review & Editing:** Mulugeta Russom.

**Writing – original draft:** Merhawi Debesai.

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
