## [Decision Letter · Decision Letter 0]

13 Jan 2020

Dear Mr. Debesai:

Thank you very much for submitting your manuscript "Praziquanel and Risk of Visual Disorders: Case Series Assessment" (PNTD-D-19-01761) for review by PLOS Neglected Tropical Diseases. Your manuscript was fully evaluated at the editorial level and by independent peer reviewers. The reviewers appreciated the attention to an important topic but identified some aspects of the manuscript that should be improved.

We therefore ask you to modify the manuscript according to the review recommendations before we can consider your manuscript for acceptance. Your revisions should address the specific points made by each reviewer.

(1) A letter containing a detailed list of your responses to the review comments and a description of the changes you have made in the manuscript.

(2) Two versions of the manuscript: one with either highlights or tracked changes denoting where the text has been changed (uploaded as a "Revised Article with Changes Highlighted" file); the other a clean version (uploaded as the article file).

(3) If available, a striking still image (a new image if one is available or an existing one from within your manuscript). If your manuscript is accepted for publication, this image may be featured on our website. Images should ideally be high resolution, eye-catching, single panel images; where one is available, please use 'add file' at the time of resubmission and select 'striking image' as the file type. 

Please provide a short caption, including credits, uploaded as a separate "Other" file. If your image is from someone other than yourself, please ensure that the artist has read and agreed to the terms and conditions of the Creative Commons Attribution License at http://journals.plos.org/plosntds/s/content-license (NOTE: we cannot publish copyrighted images). 

(4) Appropriate Figure Files 

Please remove all name and figure # text from your figure files upon submitting your revision. Please also take this time to check that your figures are of high resolution, which will improve both the editorial review process and help expedite your manuscript's publication should it be accepted. Please note that figures must have been originally created at 300dpi or higher. Do not manually increase the resolution of your files. For instructions on how to properly obtain high quality images, please review our Figure Guidelines, with examples at: http://journals.plos.org/plosntds/s/figures

While revising your submission, please upload your figure files to the Preflight Analysis and Conversion Engine (PACE) digital diagnostic tool, https://pacev2.apexcovantage.com/ PACE helps ensure that figures meet PLOS requirements. To use PACE, you must first register as a user. Then, login and navigate to the UPLOAD tab, where you will find detailed instructions on how to use the tool. If you encounter any issues or have any questions when using PACE, please email us at figures@plos.org.

We hope to receive your revised manuscript by Mar 13 2020 11:59PM. If you anticipate any delay in its return, we ask that you let us know the expected resubmission date by replying to this email.

To submit your revised files, please log in to https://www.editorialmanager.com/pntd/

Sincerely,

Xiao-Nong Zhou

Associate Editor

Hélène Carabin

Deputy Editor

Reviewer's Responses to Questions

**Key Review Criteria Required for Acceptance?**

**Methods**

-Are the objectives of the study clearly articulated with a clear testable hypothesis stated?

-Is the study design appropriate to address the stated objectives?

-Is the population clearly described and appropriate for the hypothesis being tested?

-Is the sample size sufficient to ensure adequate power to address the hypothesis being tested?

-Were correct statistical analysis used to support conclusions?

-Are there concerns about ethical or regulatory requirements being met?

Reviewer #1: Minor revision

Authors should provide a brief description of how the causality assessment was done using Austin Bradford Hills Criteria. 

Information on the type of descriptive analysis performed should be included

Reviewer #2: (No Response)

**Results**

-Does the analysis presented match the analysis plan?

-Are the results clearly and completely presented?

-Are the figures (Tables, Images) of sufficient quality for clarity?

Reviewer #1: It will be interesting to show the dose distribution of praziquantel. More so, as cases of visual impairment may be dose related.

It will be necessary to provide data on the weight distribution of the recipients with visual impairment. This is important as the authors suggested that the underweight profile of the Eritrean population may have resulted in overdosing and a possible risk factor for the occurrence of visual impairment.

Reviewer #2: (No Response)

**Conclusions**

-Are the conclusions supported by the data presented?

-Are the limitations of analysis clearly described?

-Do the authors discuss how these data can be helpful to advance our understanding of the topic under study?

-Is public health relevance addressed?

Reviewer #1: The conclusions are supported by the data presented. Authors stated the limitations of the study. Noteworthy is the fact that the data set used in this assessment comprised reports of passive surveillance. Hence, more studies are recommended to establish the dose response relationship between praziquantel and the incidence of visual impairment.

Reviewer #2: (No Response)

**Editorial and Data Presentation Modifications?**

Reviewer #1: The following aspects require minor revisions

Abstract

Pg 2 line 18- Define the abbreviations (ICSRs and ADRs) with first use 

Introduction

Pg 4 line 65- “Of note was the high proportion of visual abnormalities …” the number of cases should be included.

Reviewer #2: (No Response)

**Summary and General Comments**

Reviewer #1: This is an interesting case series assessment of the association between single dose praziquantel and reported visual impairment in an Eritrean population. The data extraction from the WHO international ICSRs Database and causality assessment were properly conducted. The authors also acknowledged the limitation of not being able to do a challenge and re-challenge as part of the causality assessment. The finding of cases of visual impairment in recipients of single dose praziquantel is important from a public health perspective and merits further investigation.

Reviewer #2: (No Response)

PLOS authors have the option to publish the peer review history of their article (what does this mean?). If published, this will include your full peer review and any attached files.

Reviewer #1: Yes: ISAAC OKOH ABAH

Reviewer #2: No

---

## [Editor Report · Decision Letter 1]

3 Feb 2020

Dear Mr. Debesai,

Thank you very much for submitting your manuscript "Praziquantel and Risk of Visual Disorders: Case Series Assessment" for consideration at PLOS Neglected Tropical Diseases. As with all papers reviewed by the journal, your manuscript was reviewed by members of the editorial board and by several independent reviewers. The reviewers appreciated the attention to an important topic. Based on the reviews, we are likely to accept this manuscript for publication, providing that you modify the manuscript according to the review recommendations. 

Sincerely,

Xiao-Nong Zhou

Associate Editor

Hélène Carabin

Deputy Editor
---

## [Editor Report · Decision Letter 2]

4 Mar 2020

Dear Mr. Debesai,

We are pleased to inform you that your manuscript 'Praziquantel and Risk of Visual Disorders: Case Series Assessment' has been provisionally accepted for publication in PLOS Neglected Tropical Diseases.

Before your manuscript can be formally accepted you will need to complete some formatting changes, which you will receive in a follow up email. A member of our team will be in touch within two working days with a set of requests.

Best regards,

Xiao-Nong Zhou

Associate Editor

Hélène Carabin

Deputy Editor

---

## [Editor Report · Acceptance letter]

7 Apr 2020

Dear Mr. Debesai,

We are delighted to inform you that your manuscript, "Praziquantel and Risk of Visual Disorders: Case Series Assessment," has been formally accepted for publication in PLOS Neglected Tropical Diseases.

Best regards,

Serap Aksoy

Editor-in-Chief

Shaden Kamhawi

Editor-in-Chief
